# Vitachelox: Protection of the Skin Against Blue Light-Induced Protein Carbonylation

**Stefano Togni [1],\*, Giada Maramaldi [1] , Andrea Cavagnino [2], Ambra Corti [3] and Luca Giacomelli [3]**

1    Indena S.p.A, 20139 Milan, Italy
2    OxiProteomics, 75005 Paris, France
3    Polistudium SRL, 20135 Milan, Italy
\*    Correspondence: stefano.togni@indena.com

**Abstract:** Protein carbonylation (PC) is a marker of reactive oxygen species-mediated alterations induced by external stimuli such as UV and blue light irradiation. In this study, we investigated the protective effect of Vitachelox®, a mixture of three standardized natural extracts rich in polyphenols, against PC induced by blue light irradiation in human keratinocytes. We tested eight experimental conditions, including Vitachelox® 0.01% and 0.005% w/v, used for 6 or 24 h before irradiation, and a solution of N-acetylcysteine (NAC) as positive control of protection. PC was evaluated by fluorescence microscopy in situ and by absolute quantification (Carbonyl Score) upon protein extraction and separation. Both the in situ visualization study and the carbonyl score showed a considerable increase in protein oxidative damage upon blue light irradiation, and a decrease in PC in the presence of Vitachelox®. In particular, Vitachelox® 0.005% showed superior results compared to NAC in terms of carbonyl score and protein quality, and it was estimated to exert a protective action against blue-light irradiation ranging from 72% (24 h) to 82% (6 h). The protective antioxidant effect of Vitachelox®, together with the anti-inflammatory and anti-microbial properties previously reported, make this natural active ingredient a valuable tool in the maintenance of healthy skin.

**Keywords:** Vitachelox®; protein carbonylation; blue light; anti-oxidant; human keratinocytes; natural extracts; polyphenols

## 1. Introduction

The skin is one of the main targets of oxidative stress, generated both by endogenous reactions and exposure to external stimuli [1]. Reactive oxygen species (ROS) are known to alter cellular structures, such as DNA, proteins, and lipids, and may lead to skin damage if not adequately controlled by the skin antioxidant defense system [2].

Protein oxidation, also known as protein carbonylation, can be caused by oxidative cleavage of proteins, direct oxidation of amino acids residues or introduction of carbonyl groups as a result of the reaction with aldehydes derived by lipid peroxidation [3]. The formation of carbonylated proteins (CPs) is recognized as a marker of ROS-mediated alterations in the skin and the presence of CPs in the stratum corneum (SC) has been associated with changes in skin features, such as mechanical properties [4] and moisture functions, including water content and trans-epidermal water loss [5]. Moreover, the accumulation of CPs seems to increase with aging and may participate in the initiation and progression of aging-related diseases, such as cardiovascular disease, cancer and inflammatory and neurodegenerative disorders [6]. Of note, CP levels appear to be increased in patients with inflammatory skin diseases associated with xerosis, such as psoriasis and atopic dermatitis, thus suggesting the implication of protein oxidative modifications in the development of inflammatory skin disorders [7].

One of the main causes of protein oxidation is exposure to sunlight, and, in particular, to the ultraviolet (UV) component of light, which is widely accepted as a contributing factor to skin damage and carcinogenesis [8]. More recently, visible blue light, which has a longer wavelength compared to UV, has also been recognized as a source of oxidative stress for skin cells, especially at a mitochondrial level [9]. Interestingly, CP formation is also influenced by seasonal changes and seems to increase during autumn and winter, thus contributing to the rough appearance of the skin in colder environments [4].

Natural products contacting polyphenols have been known for centuries for their beneficial properties and are recognized as exerting a protective action on the skin by inhibiting UV-induced inflammation, oxidative stress and DNA damage [10]. Vitachelox® is a multi-component powder composed of three standardized extracts of natural compounds rich in polyphenols: *Vitis vinifera* (grape) seeds, *Camellia sinensis* (green tea) leaves and *Quercus robur* (oak) wood/bark [11]. Grape seed extract presents antioxidative, anti-inflammatory and antimicrobial properties and seems to inhibit UV-induced tumor development thanks to active ingredients such as flavonoids, polyphenols, anthocyanins, proanthocyanidins, procyanidines and resveratrol [12]. Green tea extracts contain the catechin epigallocatechin-3-gallate (EGCG) and related compounds; EGCG is a phytochemical flavonoid with antioxidant activity that has been recently reported to decrease the photosensitivity of phospholipids to blue light oxidative damage [13]. Lastly, the extract of oak bark presents antioxidant and antimicrobial properties [14]. The physico-chemical characteristics of Vitachelox® and its ready dispersibility poses no limitations to the preparations of cosmetic formulations. Indeed, Vitachelox® is easily dispersed in water phase and incorporated into biphasic systems forming an emulsion suitable for topical application.

In this study, we investigated the protective effect of Vitachelox® on the skin by evaluating protein carbonylation in human keratinocytes after blue light irradiation.

## 2. Material and Methods

### 2.1. Experimental Design

The protective effect of Vitachelox® on protein carbonylation induced by blue light irradiation was tested in human keratinocytes (HaCat) cultured in vitro in eight different experimental conditions (Figure 1). Cells were seeded at density of 500,000 cells/well in multi-well plates, for each replicate and experimental group. Cells were incubated with a solution containing Vitachelox® at two different concentrations (0.01% and 0.005% w/v) and using two different exposure times (6 and 24 h) prior to irradiation. The two concentrations were estimated to be in the same range as the concentration obtained in the skin after topical application of Vitachelox®. A solution of N-acetylcysteine (NAC, 2.5 mM) maintained for 6 or 24 h was used as positive control of protection, while negative control consisted in irradiation of cells with no protective substance; non-irradiated cells were also used as internal control (Figure 1). For each experimental condition, we tested three replicates.

Blue light irradiation (LED source, emission peak at $\lambda = 460$ nm) was performed by using the OxiProteomics® irradiation system (OxiProteomics, Paris, France), whose main characteristics are summarized in Table 1. After irradiation, the cells were collected, snap-frozen and conserved at −80 °C for the analysis.

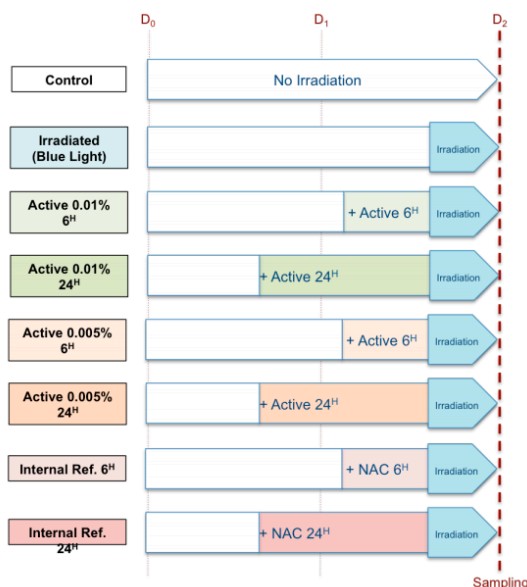

**Figure 1.** Experimental design.

**Table 1.** Main characteristics of blue light irradiation.

| Parameter | Value |
|---|---|
| Wavelength | 460 nm |
| Source | LED |
| Irradiance | 53 mW/cm$^2$ |
| Dose | 35 J/cm$^2$ |
| Irradiation time | 11 min |

*2.2. Protein Carbonylation Analysis*

Protein carbonyls assessments were performed by OxiProteomics (Paris, France). Protein carbonylation was evaluated both: in situ by fluorescence microscopy and by their absolute quantification (Carbonyl Score) upon protein extraction and electrophoresis separation.

The in-situ visualization analysis was performed by labeling carbonylated and total proteins with two different fluorescent probes. Carbonylated proteins were labeled with a specific functionalized fluorescent probe (red) [15], whereas total proteins were labeled using cyanine 3 hydroxysulfosuccinimide (Cy3NHS; green), a probe routinely used in proteomics studies. Images were collected using an epifluorescence microscope and analyzed with the software ImageJ [16]. Cells in different conditions were compared with strictly exactly exposure time, focus and resolution.

The carbonyl score was obtained for each experimental condition after protein extraction and quantification. Proteins were initially extracted by using OxiProteomics's validated protocols and quantified by the Bradford method using calibrated bovine serum albumin as standard [17]. Carbonylated proteins were labeled with specific functionalized fluorescent probes and samples were resolved by high-resolution electrophoresis separation (4–20% gradient SDS-PAGE). Total proteins were post-stained with SyproRuby™ protein gel stain. Images were acquired using the Ettan® DIGE imager (GE Healthcare, Chicago, IL, USA) and densitometric analysis of protein bands was performed by ImageJ [16]. The carbonylation score for each sample was calculated as follows:

Carbonylation score (sample x) = carbonylated protein fluorescent signal (sample x)/total protein fluorescent signal (sample x).

The average values were calculated for each experimental condition taking into consideration the replicates. Statistical analyses were conducted using GraphPad Software (GraphPad, La Jolla, CA, USA).

The protein quality index (Figure 4) was generated by the linear distribution of more than 800 internally collected data points of protein carbonylation (Carbonyl Score values from the internal database of OxiProteomics) defining a range of protein quality in function of a gradient of protein carbonylation (from low levels of carbonylation in blue to high levels in violet). The obtained average values were benchmarked against the protein quality index allowing their relative positioning in a range of protein carbonylation, beyond the single experiment results.

## 3. Results

The in-situ visualization study showed a considerable increase in protein oxidative damage upon blue light irradiation, and a decrease in protein carbonylation in the presence of Vitachelox®. As shown in Figure 2, the specific oxidative protein patterns, represented by the superposition of oxidative specific signal (red) and total protein signal (green), were different in cells not subjected to irradiation—where the green signal was dominant—compared with cells irradiated with no protective agent—where most of the cells presented a red signal, indicative of protein carbonylation. Cells treated with Vitachelox® before irradiation showed a considerable decrease in oxidative-specific fluorescence, thus suggesting a decrease in blue light-induced oxidation.

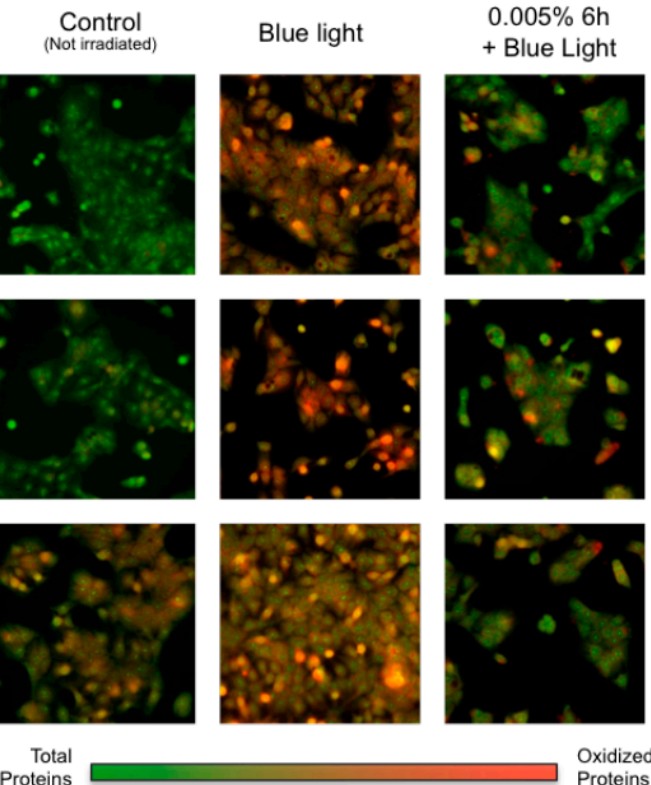

**Figure 2.** In-situ visualization of protein oxidation (red) merged with total protein signal (green).

Figure 3 shows the results of the quantification of carbonylated (Figure 3A), and total proteins (Figure 3B) and the measurement of the carbonyl score for each experimental condition (Figure 3C,D). Protein oxidative damage, as measured by the carbonyl score, was higher for irradiated cells compared to non-irradiated cells, whereas intermediate values were reported in the presence of Vitachelox® or NAC. The protective effect of Vitachelox® on protein carbonylation was visible at both concentrations and with both exposure times, although only one concentration of Vitachelox® (0.005%) reached a

statistically significant difference compared with the negative control, after 6 h of exposure. Notably, the decrease in protein damage observed with Vitachelox® 0.005% for 6 h was superior, although not statistically different, to that obtained with the positive control (NAC).

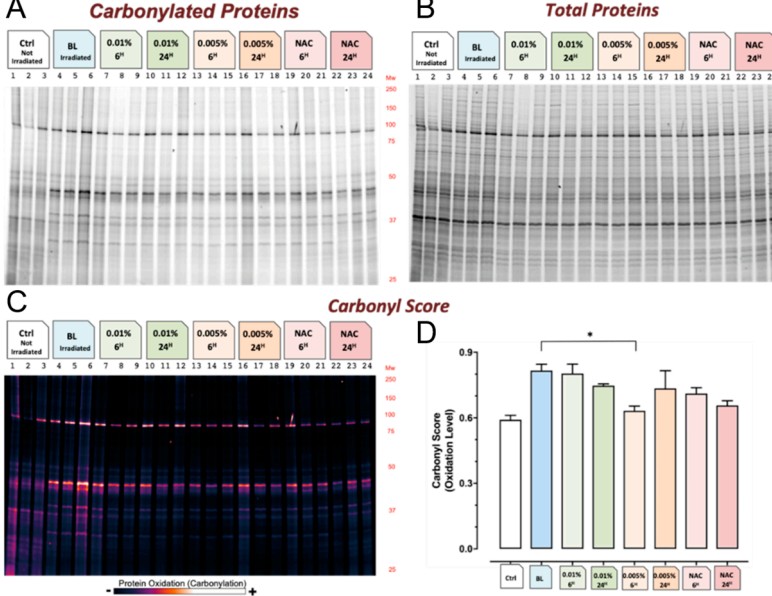

**Figure 3.** Protein quantification and carbonyl score. Quantification of carbonylated (**A**) and total (**B**) proteins. The levels of protein carbonylation for each experimental condition are represented as a continuous intensity histogram (**C**) and plotted as vertical bar representation with standard error from the average values ± standard deviation (**D**). * Significantly different from stressed experimental group ($p < 0.01$).

The carbonyl score measured in the different experimental conditions was further compared with data available in our database (Skin Protein Quality Index) to evaluate protein quality. As shown in Figure 4, a decrease in protein quality was observed upon irradiation, while a protective effect resulting in a significant improvement in protein quality was observed in cells treated with Vitachelox® or NAC. Notably, Vitachelox® 0.005% led to superior protein quality compared with the positive reference (NAC).

Finally, we measured a protection score of effect for Vitachelox® 0.005%, as this concentration reached a statistically significant protection ($p < 0.01$) from protein oxidation compared to irradiated cells. The protection score was calculated by assuming a performance of 0% for the oxidation level of the stressed group (cells irradiated with no protection) and a performance of 100% for the control group (cells not irradiated). The analysis showed that Vitachelox® 0.005% applied for 6 h had a protection score of 82%, while if applied for 24 h before irradiation the protective effect was 72% (Figure 5). Of note, when compared to the internal positive reference (NAC), a superior performance of protection was observed for both experimental conditions of Vitachelox® 0.005%.

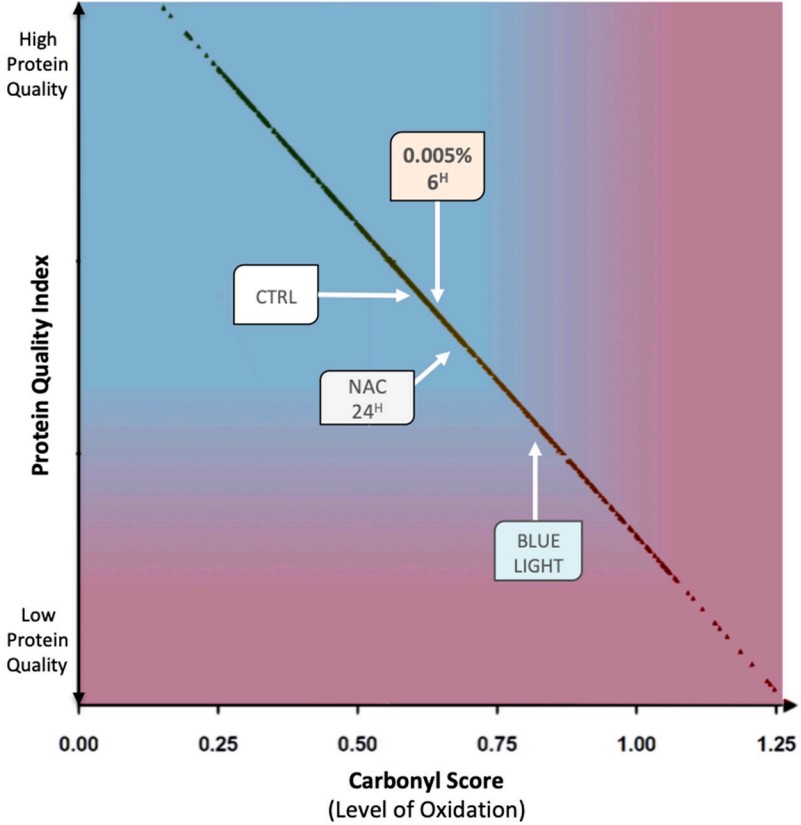

**Figure 4.** Protein quality index diagram. The average values for relevant experimental conditions are benchmarked against the protein quality index of the skin (more than 800 carbonyl score data points, black spots) allowing their relative positioning in a range of protein quality in function of a gradient of protein carbonylation (from low levels of carbonylation in blue to high levels in violet).

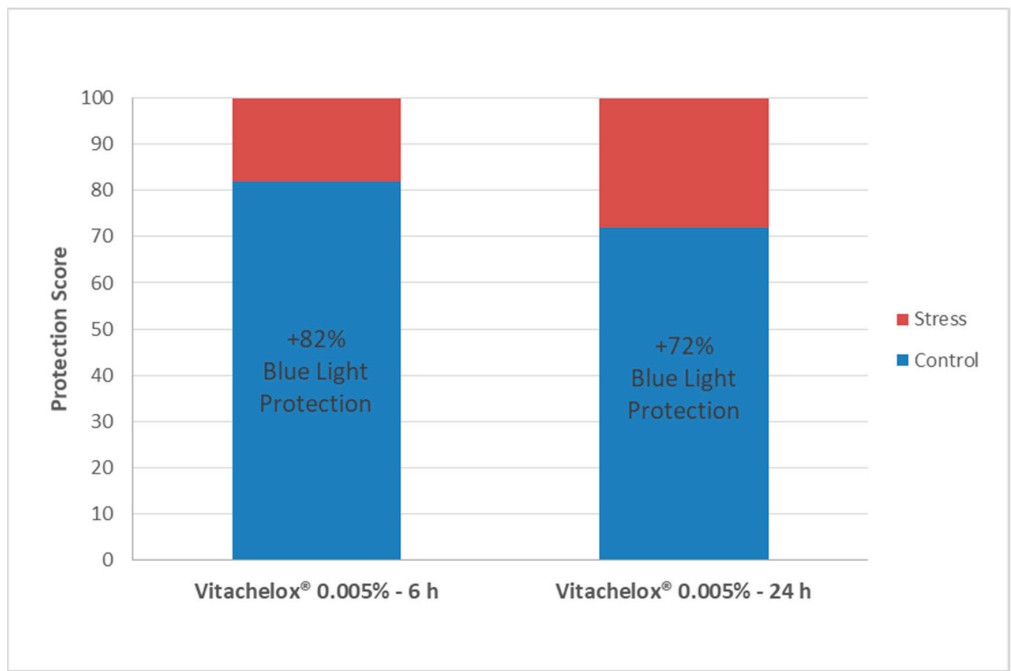

**Figure 5.** Protection Score. The values (%) represent the performance of Vitachelox® to effectively contrast the deleterious effects of blue light irradiation at a molecular level. Control refers to the not irradiated cells, while stress corresponds to cells irradiated with no protective agent.

## 4. Discussion

The results of this study show that Vitachelox® exerts a protective action on human keratinocytes, by reducing protein damage induced by blue light irradiation. This protective effect is the result of the antioxidative activity of the multiple polyphenols contained in Vitachelox®, which act by neutralizing ROS and, therefore, limit protein oxidative modifications.

As shown in the in-situ visualization study, the increase in protein carbonylation observed after cells irradiation, and suggestive of cell oxidative damage, was remarkably contained in cells treated with Vitachelox®. Similar results were obtained through the carbonyl score analysis. Indeed, a significant increase in protein oxidative damage, as measured by a higher carbonyl score, was observed upon blue light irradiation. This increase was prevented in the presence of Vitachelox® at the two concentrations tested (0.01% and 0.005% w/v). In particular, treatment with Vitachelox® 0.005% for 6 h resulted in a statistically significant protection ($p < 0.01$) from protein oxidation compared with irradiated cells, and reached superior results compared to the positive control NAC.

By limiting the oxidative damage induced by blue light irradiation, Vitachelox® seems to, therefore, contribute to the maintenance of good-quality proteins in the keratinocytes. In fact, the Skin Protein Quality Index, evaluated by comparison of current data to reference data present in our database, was higher for cells treated with Vitachelox® 0.005% compared with irradiated cells and with those treated with NAC, almost reaching the quality level of non-irradiated cells (Figure 4). Based on these results, the protective effect of Vitachelox® 0.005% against irradiation-induced protein damage is estimated to be 82% when applied on the skin for 6 h and of 72% when applied for 24 h before irradiation (Figure 5).

It is interesting to note that although we have observed a tendency to reduced protein carbonylation with Vitachelox® at different concentrations and exposure time, Vitachelox® 0.005% applied for 6 h seems to be the optimal condition for protection against CP. Since the exact mechanism of action of Vitachelox® against blue light irradiation is unknown, we cannot explain the reasons behind this observation. However, it appears that a dose/response or time-of-application/response direct correlation is not applicable in the particular conditions evaluated in this study. It is important to note that the level of protein carbonylation in a precise moment is the result of a "steady-state" status in protein homeostasis, determined by the regulation of multiple cellular mechanisms (i.e., protein turnover, ex-novo synthesis, autophagy, chaperonin-mediated protein folding and proteasome activity), in addition to external variables, such as the stress induced by irradiation and the presence of active compounds such as Vitachelox® or NAC.

The results of this study add further information to literature data on the skin-protective properties of Vitachelox®. Previous studies have already shown that this multiple component extract is able to reduce different markers of oxidative stress, such as DNA oxidation (−59% in 8-hydroxy-deoxyguanosine) and lipid peroxidation (−66% in malondialdehyde) (data on file). Moreover, Vitachelox® has been shown to protect the skin from air pollution, limiting the permeation of heavy metals—namely chromium, nickel, iron, and zinc—in the stratum corneum [11]. These properties are extremely important to maintain healthy skin and prevent skin aging but may also be relevant in the treatment of skin diseases, such as skin inflammatory conditions that seem to be associated with oxidative damage [7]. For example, a recent study suggests that Vitachelox® may contribute to the treatment of patients with acne-prone skin thanks to its antioxidant properties combined with a positive modulation of the cutaneous microflora [18].

## 5. Conclusions

Vitachelox® acts as a natural protectant active ingredient for human skin by reducing protein oxidation induced by blue light irradiation. This antioxidant effect is beneficial for the skin and adds to the anti-inflammatory and anti-microbial properties of Vitachelox®, making it a valuable tool in the maintenance of healthy skin and possibly a useful contribution to the treatment of different skin conditions.

**Author Contributions:** S.T., G.M. and A.C. (Andrea Cavagnino) have contributed to the conception and design of the work, acquisition, analysis, and interpretation of data; A.C. (Ambra Corti) and L.G. have contributed to the interpretation of data, and drafted the work. All the authors have revised the work, have approved the submitted version and agree to be personally accountable for the their contributions and for ensuring that questions related to the accuracy or integrity of any part of the work, even ones in which they were not personally involved, are appropriately investigated, resolved, and documented in the literature.

**Funding:** No funding related to this study was provided.

**Acknowledgments:** The authors thank Aashni Shah (Polistudium srl) for editorial assistance, which was supported by internal funds.

**Conflicts of Interest:** S.T. and G.M. are Indena employees. A.C. (Andrea Cavagnino) is OxiProteomics employee. The other Authors declare no conflict of interest directly relevant to this study.

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
