# Peer review of "Vitachelox: Protection of the Skin Against Blue Light-Induced Protein Carbonylation"

_cosmetics, doi:10.3390/cosmetics6030049_

Round 1

Reviewer 1 Report

The paper by Togni et al. entitled "Vitachelox: protection of the skin against blue light-induced protein carbonylation" is a research paper examining the protective effect of Vitachelox against blue light damage on keratinocytes. Overall, the paper is well written and structured. 

I would like to raise the following concerns.

1, The authors showed anti-phototoxic effect of Vitachelox (0.01% and 0.005%), but it is difficult to interpret it.  The authors should show why the authors choose the concentration of 0.01% and 0.005%. It is better to show the concentration of the Vitachelox in human real skin when the VItachelox is administered at the recommended dose.

2, Additional information about the Vitachelox is needed. Most of the readers will be unfamiliar to Vitachelox. Is it oral use? or topcal?

3, Introduction section, Lines 39-43 (Moreover...disorders[7]). I don't agree with the statements. The sentences are very odd to me (dermatologist). Please reconsider the sentences.

Author Response

Dear Editor,

Please find enclosed a revised version of our manuscript entitled “Vitachelox: protection of the skin against blue light-induced protein carbonylation”.

We are thankful to the reviewers for their comments, and we appreciate the suggestions and criticisms that in fact helped us improve the manuscript. Our responses to the reviewers’ criticisms are reported below. The text and some of the figures have been modified accordingly, as detailed below.

Best regards

Stefano Togni

Reviewer #1

The paper by Togni et al. entitled "Vitachelox: protection of the skin against blue light-induced protein carbonylation" is a research paper examining the protective effect of Vitachelox against blue light damage on keratinocytes. Overall, the paper is well written and structured.

Authors’ Response: Thank you for your comments and your consideration of our work.

I would like to raise the following concerns.

1) The authors showed anti-phototoxic effect of Vitachelox (0.01% and 0.005%), but it is difficult to interpret it. The authors should show why the authors choose the concentration of 0.01% and 0.005%. It is better to show the concentration of the Vitachelox in human real skin when the Vitachelox is administered at the recommended dose.

Authors’ Response: The two concentrations have been chosen based on a previous study in which a similar product was tested in the peroxidation of food matrix. We estimated that the two concentrations used are in a similar range as those obtained with the topical administration of Vitachelox®. This information has been added in the manuscript.

2) Additional information about the Vitachelox is needed. Most of the readers will be unfamiliar to Vitachelox. Is it oral use? or topical?

Authors’ Response: Thank you for your suggestion; we added more information on Vitachelox® formulation and its use as a topical application.

3) Introduction section, Lines 39-43 (Moreover...disorders[7]). I don't agree with the statements. The sentences are very odd to me (dermatologist). Please reconsider the sentences.

Authors’ Response: We have modified the sentence to clarify the meaning.

Reviewer 2 Report

The experiments exposed in the article seem convincing of the protective effect of Vitachelox against protein carbonylation. However often (not always) the results are exposed in a hasty and superficial way and more details should be provided to complete and improve the quality of the information and make the results more convincing. Some minor revisions must be made to ensure the quality of the publication.

1) What kind of cells are you using during the experiments? In the experimental design section the authors say they conducted their experiments on human keratinocytes. Are they primary cultures of keratoniocytes or a cell line? In the latter case, which one?  

2) what concentration of NAC was used in control experiments?  

3) Were the cells all plated to the same density during different treatments / experiments? In figure 2 the cells appear to have been plated at different densities (especially those exposed to vitachelox and blue light) and the different cellular desity can lead to a different response to the treatments.
Has cell viability / growth been assessed (here or in another work that should still be mentioned) in the presence of different treatments? Vitachelox does not alter cell viability / growth? Data concerning cell viability / growth should be provided under the experimental conditions used.  

4) a scale bar should be placed on the photos (at least one)  

5) how many replicates have been made for gel quantification of carbonylated proteins? if only 3 how could you establish that line 16 had an out of range value (quoting line 133: due to its aberrant behavior (difference from standard deviation)). From a statistical point of view it makes no sense ... can you explain it better? moreover even during the treatment with vitachelox 0.01 for 6 hours we observe an aberrant datum of the same type, but you have not excluded it, why? Including the data in line 16 would probably have a comparable value.
In conclusion, if only 3 replicates have been made, the same number should be repeated to make the result more convincing

6) graph 4: how was it obtained? what are the values reported in the y axis? there are no values, there is no legend, the graph is illegible and cannot be interpreted. there seems to be a regression line, obtained in what way? Further information on how it was obtained and on the values (type, unit of measurement) reported on the y axis must be provided.

7) why does a lower concentration of vitachelox (and for less time, 6 h compared to 24 hours) seem to have a greater protective effect? Authors should discuss the data in more detail, especially for future practical applications of the product. Is it possible that the Vitachelox has a hormetic effect as already reported for several and numerous phytochemicals and other pharmacological compounds? In such a case a more complete dose response curve should be provided before its application and the effects should be carefully studied in vivo.

Carbonylation is a generic term which refers to reactive carbonyl groups present in biomolecules due to oxidative reactions induced by reactive oxygen species not oxigen as stated in the line 10 of the abstract.

Author Response

Dear Editor,

Please find enclosed a revised version of our manuscript entitled “Vitachelox: protection of the skin against blue light-induced protein carbonylation”.

We are thankful to the reviewers for their comments, and we appreciate the suggestions and criticisms that in fact helped us improve the manuscript. Our responses to the reviewers’ criticisms are reported below. The text and some of the figures have been modified accordingly, as detailed below.

Best regards

Stefano Togni

Reviewer #2

The experiments exposed in the article seem convincing of the protective effect of Vitachelox against protein carbonylation. However often (not always) the results are exposed in a hasty and superficial way and more details should be provided to complete and improve the quality of the information and make the results more convincing. Some minor revisions must be made to ensure the quality of the publication.

Authors’ Response: Thank you for revising the manuscript and for your useful suggestion. We appreciate your comments that have helped us improve the quality of the information provided.

1) What kind of cells are you using during the experiments? In the experimental design section the authors say they conducted their experiments on human keratinocytes. Are they primary cultures of keratoniocytes or a cell line? In the latter case, which one?

Authors’ Response: We added more information on the cells used (HaCaT cell line).

2) What concentration of NAC was used in control experiments?

Authors’ Response: NAC was used at the concentration 2.5 mM. We added this information in the manuscript.

3) Were the cells all plated to the same density during different treatments/experiments? In figure 2 the cells appear to have been plated at different densities (especially those exposed to vitachelox and blue light) and the different cellular density can lead to a different response to the treatments.

Has cell viability/growth been assessed (here or in another work that should still be mentioned) in the presence of different treatments? Vitachelox does not alter cell viability/growth? Data concerning cell viability/growth should be provided under the experimental conditions used.

Authors’ Response: Cells were seeded at a density of 500,000 cells/well in multi-well plates, for each replicate and experimental group. This information has been added to the manuscript.

Although the cells were seeded at the same density in each replicate and experimental condition, the distribution of cellular density was more variable in the control group and in the stressed group (irradiated cells with no protective agent) compared to cells treated with Vitachelox® 0.005% for 6 hours, where the distribution between the replicates was more homogeneous. However, the relative increase in carbonylation after irradiation was homogeneous in each replicate, as well as the decrease in protein carbonylation observed in the Vitachelox®/NAC groups compared to the stress group. The results of the in situ visualization are therefore in line with the quantification of carbonylation.

We have not included a cellular viability test in our study or in other studies, however before irradiation we have performed a qualitative check of cellular status by optical microscopy: these observations excluded any toxic effect of Vitachelox® on cellular phenotype (morphology).

4) A scale bar should be placed on the photos (at least one)

Authors’ Response: Unfortunately it is not possible to add a scale bar as this is a qualitative evaluation, as shown in the bar below the photos.

5) How many replicates have been made for gel quantification of carbonylated proteins? if only 3 how could you establish that line 16 had an out of range value (quoting line 133: due to its aberrant behavior (difference from standard deviation)). From a statistical point of view it makes no sense ... can you explain it better? moreover even during the treatment with vitachelox 0.01 for 6 hours we observe an aberrant datum of the same type, but you have not excluded it, why? Including the data in line 16 would probably have a comparable value.

In conclusion, if only 3 replicates have been made, the same number should be repeated to make the result more convincing

Authors’ Response: Thank you for your detailed comment and suggestion. For each experimental condition we have performed 3 replicates, as now mentioned in the manuscript.

Following your suggestion we have rerun the statistical analysis including all 3 replicates for Vitachelox® 0.005% - 24h and updated Figure 2 and the results section accordingly. Based on the new statistical analysis the performance observed for Vitachelox® 0.005% - 24h is increased from 72% to 73% and loses its statistical significance compared to the stress group. However the difference between cells irradiated with no protective agent and Vitachelox® 0.005% - 6h maintains its significance (p<0.05).

The increase in protein carbonylation after irradiation is equal to 38%, which is significantly different compared to the control group (p<0.01). We have also observed a tendency to reduced protein carbonylation with Vitachelox® (31% for 0.001% - 24h, 82% for 0.005% - 6h, and 37% for 0.005% - 24h) although not all the values reached statistical significance. With reference to NAC, the protective effect is equal to 48% and 71%, respectively for 6 and 24 h application, with the second one being statistically superior to control.

Although the suggestion of the reviewer is valid we would like to clarify more in details the reason why we had decided to exclude the highest value for the experimental condition Vitachelox® 0.005% - 24h. The value was considered as aberrant, and therefore excluded from the data analysis as, based on the interpretation of the whole results, if this data was included the standard deviation would have been 0.141, equal to the 63% differential between stress and control (0.225). This value is higher compared to those registered in all the other experimental conditions and almost double compared to the other experimental condition mentioned by the reviewer (33% Vitachelox® 0.01% - 6h). For reference the other values are 16% (control); 22% (stress); 6% (Vitachelox® 0.01% - 24h); 17% (Vitachelox® 0.005% - 6h); 21% (NAC 6h) and 17% (NAC - 24h). Moreover, in this replicate the difference between median and maximum value was the highest compared to other experimental conditions (0.205 for Vitachelox® 0.005% - 6h; 91% of the differential stress-control vs 0.124 for Vitachelox® 0.001% -6h, 55% differential). As a reference the other differentials are 70% for Vitachelox® 0.005% - 6 h and 38% for Vitachelox® 0.01% – 6 h. Based on these observations we had decided to exclude the replicate with the highest value from data analysis, considering it as an aberrant value.

6) Graph 4: how was it obtained? what are the values reported in the y axis? there are no values, there is no legend, the graph is illegible and cannot be interpreted. There seems to be a regression line, obtained in what way? Further information on how it was obtained and on the values (type, unit of measurement) reported on the y axis must be provided.

Authors’ Response: We have added more information on Figure 4 including y axis labeling and a detailed explanation of how the Protein Quality Index has been generated.

7) Why does a lower concentration of vitachelox (and for less time, 6 h compared to 24 hours) seem to have a greater protective effect? Authors should discuss the data in more detail, especially for future practical applications of the product. Is it possible that the Vitachelox has a hormetic effect as already reported for several and numerous phytochemicals and other pharmacological compounds? In such a case a more complete dose response curve should be provided before its application and the effects should be carefully studied in vivo.

Authors’ Response: Thank you for your interesting comment, we have further discussed the results of the study here below and in the discussion section.

As we do not know the mechanism of action of Vitachelox® on this particular type of stress we cannot further interpreter the efficacy results observed in the study. However, it is worth mentioning that also in the past we observed an increased protective effect associated with active compounds used for less time or at lower concentrations. These observations suggest that there may not be a dose-response or time of application-response direct correlation, at least for the experimental conditions analyzed in this study

It is important to note that the level of protein carbonylation in a precise moment is the result of a “steady-state” status in protein homeostasis, determined by the regulation of multiple cellular mechanisms (i.e. protein turn-over, ex-novo synthesis, autophagy, chaperonin-mediated protein folding and proteasome activity), in addition to external variables such as the stress induced by irradiation and the presence of active compounds such as Vitachelox® or NAC.

With reference to the experimental condition tested in this study, we have observed a tendency to reduced protein carbonylation with Vitachelox® compared to cells irradiated with no active compounds (31% for 0.001% - 24h, 82% for 0.005% - 6h, and 37% for 0.005% - 24h) although not all the values reached statistical significance. The experimental condition of Vitachelox® 0.005% - 6h appears to be the optimal condition among those tested.

Carbonylation is a generic term which refers to reactive carbonyl groups present in biomolecules due to oxidative reactions induced by reactive oxygen species not oxygen.

Authors’ Response: Thank you for pointing this out. We have revised the whole text to correct oxygen into reactive oxygen species.